# Exploring Semi-Quantitative Metagenomic Studies Using Oxford Nanopore Sequencing: A Computational and Experimental Protocol

**DOI:** 10.3390/genes12101496

**Published:** 2021-09-25

**Authors:** Rohia Alili, Eugeni Belda, Phuong Le, Thierry Wirth, Jean-Daniel Zucker, Edi Prifti, Karine Clément

**Affiliations:** 1École Pratique des Hautes Études, PSL University, Les Patios Saint-Jacques, 4-14 Rue Ferrus, 75014 Paris, France; rohia.alili@aphp.fr (R.A.); thierry.wirth@mnhn.fr (T.W.); karine.clement@inserm.fr (K.C.); 2Nutrition Department, CRNH, Assistance Publique Hôpitaux de Paris, Pitié-Salpêtrière Hospital, 75013 Paris, France; 3Nutrition and Obesity, Systemic Approaches (NutriOmics), INSERM, Sorbonne Université, 75013 Paris, France; phuongleee@gmail.com (P.L.); jdzucker@gmail.com (J.-D.Z.); edi.prifti@ird.fr (E.P.); 4Unit of Insect Vector Genetics and Genomics, Integrative Phenomics, 8 Rue des Pirogues de Bercy, 75012 Paris, France; 5Département Systématique et Evolution 16 Rue Buffon, ISYEB, UMR-CNRS, 75231 Paris, France; 6Unité de Modélisation Mathématique et Informatique des Systèmes Complexes, Institute of Research for Development(IRD), Sorbonne Université, 93143 Bondy, France

**Keywords:** semi-quantitative metagenomics, microbiome, obesity, gut microbiota, microbial DNA extraction, sequencing, simulation, Oxford Nanopore Technologies, MinION

## Abstract

The gut microbiome plays a major role in chronic diseases, of which several are characterized by an altered composition and diversity of bacterial communities. Large-scale sequencing projects allowed for characterizing the perturbations of these communities. However, translating these discoveries into clinical applications remains a challenge. To facilitate routine implementation of microbiome profiling in clinical settings, portable, real-time, and low-cost sequencing technologies are needed. Here, we propose a computational and experimental protocol for whole-genome semi-quantitative metagenomic studies of human gut microbiome with Oxford Nanopore sequencing technology (ONT) that could be applied to other microbial ecosystems. We developed a bioinformatics protocol to analyze ONT sequences taxonomically and functionally and optimized preanalytic protocols, including stool collection and DNA extraction methods to maximize read length. This is a critical parameter for the sequence alignment and classification. Our protocol was evaluated using simulations of metagenomic communities, which reflect naturally occurring compositional variations. Next, we validated both protocols using stool samples from a bariatric surgery cohort, sequenced with ONT, Illumina, and SOLiD technologies. Results revealed similar diversity and microbial composition profiles. This protocol can be implemented in a clinical or research setting, bringing rapid personalized whole-genome profiling of target microbiome species.

## 1. Introduction

In recent years, there has been a burst in knowledge related to gut microbiota screening in chronic diseases. The increasing access to high-throughput sequencing has led to the discovery of alterations in the composition of intestinal microbiota in many human disorders, including metabolic diseases. Currently, there is a real challenge to discover reproducible gut microbial signatures for diseases in order to develop generalizable diagnostic and prognostic tools, which makes their clinical use difficult [1].

In the field of metabolic diseases, microbial diversity is generally representative of microbiome and host health, as exemplified in previous studies, such as MetaHIT [2], HMP [3], MetaCardis [3] and others covering severe obesity, bariatric surgery [4], diabetes, NAFLD/NASH [5], and cirrhosis [6,7]. In mild [8] and severe obesity [9], for instance, we previously showed that reduced microbial gene richness linked to altered composition was found in 40% to 75% of the subjects and was associated with a more deleterious host phenotype. Even with these established signatures in metabolic diseases, the gut microbiome varies greatly in composition and abundance from one individual to another. 

Presently, most microbiome research studies are carried out using 16S ribosomal RNA genes or whole-genome shotgun sequencing (WGS), the latter requiring extensive computational resources and pipelines. Moreover, not all medical and research centers are able to set up high-end shotgun sequencing platforms due to multiple constraints. As opposed to previously existing technologies, Oxford Nanopore Technologies (ONT) proposes real-time sequence data generation with fewer resources and a small benchtop footprint. In the context of metagenomics, the long reads generated by ONT have led to major improvements in the de novo assembly of microbial genomes from metagenomic samples [10,11]. It has been applied to target pathogen and viral profiling [12,13], as well as the characterization of microbial communities in diverse environments from 16S data [14]. However, there is a need to define standardized wet-lab and bioinformatics protocols for the use of ONT in large-scale quantitative metagenomic studies given that most of the quantitative metagenomic bioinformatics pipelines are adapted to short reads [14]. 

In addition to biological variation, gut microbiome quantification is subject to technical variation along the preanalytical process, including sample collection and extending to DNA extraction, library preparation, and sequencing but also along the bioinformatics analytical protocols [15]. This is observed in the literature with frequently non reproducible results [16,17], highlighting the need for technical standardization [18]. Even though progress has been made with the work of different international consortia [19,20] to standardize protocols, there is still a need for fast-track and affordable microbiome screening protocols in clinical settings. For example, among critical steps prior to sequencing is DNA extraction. Costea et al. [20] reported variability in microbial composition and diversity with different DNA extraction protocols. Extraction protocols, with or without bead-beating, increase the representation of Gram-positive bacteria, as is also the case for different DNA extraction kits: the richness is higher and reads are longer with the Qiagen compared with Magnapure kits [21]. Library preparation has also an impact on the relative abundances of taxonomic and functional microbial objects [10]. Finally, the bioinformatics pipelines can yield consequent variability in microbial ecosystem description [22].

Here, we have explored protocols with the quest to optimize ONT for microbiome analyses and have proposed a complete protocol, including wet-lab preparation (i.e., sample collection, DNA extraction, and library preparation), as well as data processing and analysis. In particular, we have set up a customized analytical pipeline to estimate microbial composition and diversity, as well as to classify ONT reads using the latest bacterial gene catalogs along with functional profiling. This protocol is open access, allowing for replication and implementation within the world’s medical or research centers (https://git.ummisco.fr/pipelines/nanopore, accessed on 22 September 2021).

## 2. Materials and Methods

### 2.1. Study Design

To determine the optimal parameters for semi-quantitative estimation of microbiome ecosystems, we first optimized our bioinformatics pipeline based on controlled experiments of simulated data [23]. We simulated sequence data based on a set of known bacterial genomes and abundance distribution profiles similar to real metagenomics varied in terms of composition, richness, and sequencing depth. The simulator took into account the particularity and biases of ONT sequences. Next, we built and adapted a bioinformatics pipeline, and we searched for the best hyper parameters to minimize the difference between the estimated quantified features (abundance, richness) and the real abundance used to parameterize the simulation (Figure 1a). 

In addition, we conducted multiple wet-lab experiments to establish an optimized preanalytical protocol, from stool collection and DNA extraction and fragmentation to end-repair steps (Figure 1b). Finally, we validated our protocol and pipeline using human stool samples sequenced with different technologies (ONT, Illumina, and SOLiD) (Figure 1c).

### 2.2. ONT Microbiome-Like Simulated Data 

We set up a data simulation framework to estimate the performance of the quantification pipeline, while maximizing representation to real human gut microbial ecosystems. We used 506 reference genomes included in the construction of the Integrated Gene Catalog of human gut (IGC) [24]. We simulated 10 samples (M1:M10) whose abundances followed a Pareto distribution estimated using real metagenomic profiles of metagenomic species [9,25] computed on the same IGC [24]. We included two important variables for the quantification of microbial ecosystems into the simulation: richness (number of present species) and sequencing depth (i.e., number of reads generated by the sequencing). We simulated the variation in richness from 50 to 450 species (R50:R450), as well as the sequencing depth ranging from 1× to 5×, the complete coverage of the genomes present. In total, 250 samples were simulated using the CAMISIM software (option: NanoSim tool with default error parameters for the *Escherichia. coli* example) [26] (Figure 1a). 

### 2.3. Bioinformatics Workflow for Taxonomic Binning of ONT Sequencing

The proposed bioinformatics workflow for semi-quantitative metagenomic (QM) analyses from ONT shotgun sequencing starts with fast5 files generated by the MiniKNOW^TM^ software. The first step of the workflow consists of base calling and demultiplexing the fast5 files into fastq files. Here we used Albacore (v2.1.10) and Guppy (v2.1.3) ONT base callers from the community site available to ONT customers [27] together with custom R scripts that parse the sequence_summary.txt files, generated during the base-calling step, to generate different visualizations of the quality of the sequencing (active channels in the flow cell, distribution of active channels through time, yield in terms of reads of the run, and read length distribution). 

The taxonomic binning of ONT reads is carried out using two different reference resources. 

Centrifuge-based taxonomic binning: Centrifuge [28] was used for the taxonomic binning of individual ONT reads using their comprehensive reference database of more than 8000 reference genomes from prokaryotes and viruses (including human reference genome). This step allows for excluding human sequence reads. To remove spurious taxonomic assignments, we additionally mapped a read bin product of the initial Centrifuge classification against the corresponding reference genome from the centrifuge database using minimap2 with the map-ont option optimized for ONT reads [29]. Based on simulation experiment results, only sequences with a minimum mapQ score of 5 were retained for subsequent analyses (see results). A species relative abundance table was generated by summing the counts of each taxonomic bin (NCBI taxonomy identifiers) from the filtered Centrifuge results. This relative abundance table was combined with the experiment metadata information and a reference taxonomic table reconstructed from Centrifuge NCBI taxonomy identifiers using the R package taxize v0.9.95 [30] using phyloseq v1.30.0 [31], generating a phyloseq-class R object. This object can be used for microbial ecology analyses (rarefaction, alpha-diversity, beta-diversity, and differential abundance analysis).IGC-based taxonomic binning: A complementary approach consisting of quantifying the abundance of microbial genes. Here, ONT reads were aligned against the Integrated Gene Catalog of reference genes of the human gut microbiome (IGC) catalog [24] using minimap2 with the map-ont option [29]. The alignment of long ONT reads over short or fragmented IGC genes provided two different types of multiple mappings (an ONT read mapped over several genes). First, a long ONT read could cover a genomic region harboring more than one gene, so different genes can be mapped over nonoverlapping regions of an ONT read, providing a structural annotation of the corresponding DNA region. Second, multiple genes can also be mapped in overlapping regions of a read. These second multiple mappings were filtered out using the GenomicRanges and plyrRangesR packages [32,33], allowing for retaining the genes with the highest mapQ score and sequence identity across each alignment region. The raw gene abundance table was reconstructed by counting the number of times each gene was mapped by ONT reads. From this gene count table, the abundance of metagenomic species (MGS; coabundant gene groups clustered from 1267 human gut metagenomes used to construct the IGC [25]) was estimated as the mean value of the 50 most connected genes in each MGS as proposed in the original study [25].

### 2.4. Bioinformatics Workflow for Functional Profiling of ONT Sequencing

The final step of the bioinformatics workflow consisted in the quantification of KEGG orthology groups (KO groups) [34]. KO abundances were quantified from the results of both taxonomic binning approaches. From IGC abundance tables, KO abundances were quantified using available reference annotation from the IGC as the sum of the individual abundances of genes annotated with different KO groups [24]. For taxonomic results produced from Centrifuge quantification, we retrieved the KO content of KEGG genomes from the KEGG API [35] for which species-level pan-genomes were reconstructed for all species-level bins based on NCBI taxonomy and matched with genomic sequences in the Centrifuge database. Based on this matching, the abundance of KO groups from Centrifuge results was computed as the sum of the abundances of the species containing these KO groups. The pan-genome strategy fits with the compressed nature of Centrifuge genomes at the species level, followed to reduce the size of the indexes and improve the overall performance of the classification process [28].

### 2.5. Study Participants for Wet-Lab Experiments

Stool samples used for wet-lab protocol optimization were collected from healthy French volunteers (*n* = 15; men = 8; BMI, 18–25 kg/m^2^) from the European “MetaCardis” cohort [3]. For the comparison between sequencing technologies, we used 33 baseline samples from the Microbaria study [9], where the gut microbiome of subjects with severe obesity was characterized before and after bariatric surgery [9]. 

### 2.6. Sample Collection and Bacterial DNA Extraction for Preanalytic Protocol Experiments: Fresh Stools Were Collected with Two Different Methods

(1) A dry spoon tube (SARSTED) requires storage at −80 °C, and (2) a tube containing DNA/RNA stabilizing solution can be kept at room temperature, −20 °C, or 80 °C depending on the storage duration. For the latter collection method, we tested three available commercial kits, including (1) “DNA/RNA Shield-Fecal Collection Tube” (Zymo marketed by Ozyme), (2) “Stool Nucleic Acid Collection and Preservation Tubes” (Norgen Biotek), and (3) “Omnigen Gut for Microbiome” (DNA Genotek). 

To extract bacterial DNA, we tested four different commercial kits using manual extraction protocols: (1) “PureLink™ Microbiome DNA Purification Kit” (Invitrogen, Paris, France), (2) “Qiamp PowerFecal DNA Kit” (Qiagen, Courtaboeuf, France), (3) “ZimoBiomics DNA Mini Kit” (Ozyme Saint-Cyr-l’École, France), and (4) “Power Microbiome RNA/DNA Isolation Kit” (Mo Bio, Courtaboeuf, France). We used the “Maxwell Instrument,” a robotic station from Promega that extracts DNA from 16 samples simultaneously. We also tested automated extraction with two different kits: “Maxwell RSC Buffy Coat DNA Kit” (Promega 1, Charbonnières-les-Bains, France) and “Maxwell RSC PureFood GMO and Authentication Kit” (Promega 2, Charbonnières-les-Bains, France). Extracted stool DNA yield and quality were evaluated with a fluorometer (Qubit, Life Technologies Alfortville, France) and NanoDrop (Thermo Scientific, Alfortville, France), respectively. 

### 2.7. Optimization of DNA Extraction, DNA Fragmentation, and End Repair

DNA extraction tests were performed from stool samples collected in dry tubes from three healthy subjects from the MetaCardis cohort (BMI <25 kg/m^2^) at three sampling times for subject 01. After collection, stool samples were aliquoted and immediately stored at −80 °C. Each sample was extracted according to the protocols proposed by the manufacturer. After extraction, the samples were evaluated using the “Qubit” fluorometer to estimate the DNA yield obtained in ng/µl and using NanoDrop to evaluate DNA quality. 

### 2.8. Library Preparation and Sequencing

We used 1.5 µg of DNA to perform the library construction. Extracted DNA was fragmented in g-tubes from Covaris, and DNA end repair was performed using the NEBNext FFPE Repair Mix from New England Biolabs (NEB). We used the NEBNext Ultra II End Repair/dA-Tailing Module (NEB) for the “end prep” step, 1D Native Barcoding Genomic DNA Kit (ONT), and “NEB Blunt/TA Ligase Master Mix kit” (NEB) for DNA multiplexing and adapter ligation. We used Agencourt AMPure XP (Beckman Coulter) beads for DNA purification.

Whole-genome metagenomic sequencing was performed with ONT’s MinION tool using flow cells on which 12 samples were simultaneously loaded per run. A total of 33 samples from the Microbaria study were sequenced in parallel with ONT and Illumina NovaSeq (2 × 150 bp PE reads). Illumina sequences were processed following the same procedure as described in the original Microbaria study [9] in order to estimate microbial gene richness and the abundances of metagenomic species based on the 9.9 million genes in the IGC [24].

### 2.9. Statistical–Ecological Analyses

All statistical analyses were performed on R v.3.6. Wilcoxon rank-sum tests (for two-level categorical variables), and Kruskal–Wallis tests (for categorical variables with more than two levels) were used to compare differences in microbial diversity between experimental conditions in different experiments. *p*-Values < 0.05 (alpha level) were considered significant. Spearman correlation tests were used to compare the abundance of taxonomic and functional features between sequencing technologies (SOLiD, Illumina, Nanopore) in Microbaria samples, followed by the correction for multiple comparisons with the Benjamini–Hochberg method. Adjusted *p*-values < 0.05 were considered significant. 

Raw abundance table products of ONT sequencing were rarefied to the minimum sequencing depth in each experiment before ecological analyses. Permutational analyses of variance (PERMANOVA) with the adonis function of vegan R package [36] were used to evaluate the impact of different covariates on microbiome composition in different experiments using a Bray–Curtis beta-diversity dissimilarity matrix computed from genus-level abundance data. Alpha-diversity was estimated with phyloseq v1.30.0 [31].

## 3. Results

### 3.1. Metagenome Simulations Identified Key Pipeline Parameters for ONT Microbiome Quantification

The metagenome simulation approach allowed for evaluating the impact of different steps and parameters in the bioinformatics pipeline. The evaluation accuracy consisted of comparing the estimated abundance of microbial features (i.e., species abundance) with the original values used to generate the sequences of over 100 million long reads for 250 simulated metagenomes (see methods; additional files in Appendix A). These reads were aligned against the 506 reference genome catalog using a minimap2 aligner with the map-ont configuration, designed for optimal performance and accuracy with ONT sequencing data [29]. On average, 381,000 reads per sample (94%) were aligned against the reference genomes. The reads that could not be aligned were on average 2.5 times shorter in size (average read length = 3168 bp, sd = 24) compared with those that could (average read length = 8064 bp, sd = 8) (Figure 2a, Appendix A), suggesting that read length is a key parameter. 

We evaluated the accuracy of the estimated species abundance and richness to the reference values used for the simulation based on filtering using a primary alignment parameter (PA), defined as the best alignment of a single read among all possible multiple alignments. We compared the species abundance and richness of PA-filtered minimap2 results with those obtained without applying the PA-filtering step (raw abundances/richness).

For species richness, we observed that raw quantifications detected all reference species in simulated samples (100% samples with recall values equal to 1; Figure 2b). Quantifications based on PA-filtered reads detected all reference species (recall values equal to 1) in 95.8% of the simulated samples, with missing species observed in 21 simulations across different community compositions with low simulated sequencing depth (2 samples with 2× simulated depth of reference genomes; 19 with 1× sequencing depth; Figure 2b). However, both raw and PA-filtered reads overestimated the number of species especially in the low-richness community compositions. For all community compositions, taxonomic profiles from PA-filtered data reached higher precision values in species richness estimates compared with raw alignments (Figure 2c). Importantly, principal coordinates analysis (PCoA) using a Bray–Curtis beta-diversity dissimilarity matrix showed that PA-filtered samples were more similar to the corresponding reference distributions (R^2^ effect size dissimilarities = 0.04, *p*-value = 0.001; PERMANOVA test) compared with raw alignment samples (R^2^ effect size dissimilarities = 0.51, *p*-value = 0.001; PERMANOVA test) (Figure 2d). This suggests that the noise introduced by secondary alignments significantly decreased the precision of the pipeline.

### 3.2. Alignment Identity and Alignment Quality Affect Workflow Precision

We next evaluated the impact of filtering read alignments at different thresholds of sequence identity on the accuracy of the estimated microbiome profiles. The recall values were close to 1 for species richness when filtering by identity levels up to 40%. This means that all reference species in each simulated sample were detected by the workflow. When progressively increasing identity levels from 50% to 90%, the fraction of reference species not detected notably decreased (Appendix A). This resulted, however, in the increase in the precision of the estimated richness as the filtering lowered the number of false positives (Appendix A). At the level of similarities between relative abundance estimates, the Spearman rhos of the correlation between the estimated species abundance and the reference values decreased as the alignment identity threshold increased across all different community compositions in a similar way as recall values, showing that the loss of species with high stringent identity thresholds leads to a decrease in the overall similarities of simulated relative abundance profiles with the reference abundances (Appendix A). This has an impact on the overall microbiome composition similarities based on ordination framework. When considering the overall microbial composition, the higher was the stringency of the alignment identity, the more dissimilar the metagenomic profiles were from the reference composition of simulated samples (Appendix A), despite the presence of false positives. Overall, these results showed that common approaches to filter read alignments used in the context of second-generation sequencing technologies (NGS) (e.g., identity thresholds above 80–90% sequence identity) were not directly applicable to high error-prone ONT sequencing data. Additional parameters were needed to be explored in order to improve the accuracy of the resulting metagenomic profiles.

Thus, we next explored the mapping quality score (mapQ) as a parameter for filtering ONT sequence alignments. MapQ, as computed by minimap2, assigns high values to long reads and for which the scores assigned to secondary alignments were weak when compared with primary alignments [29]. A total of 11 different mapQ thresholds (from 0 to 50 at steps of 5) were evaluated based on the primary alignment of simulated datasets (Appendix A). In terms of recall in species richness estimates, we observed a similar decrease with the stringency of the mapQ threshold as observed with the alignment identity, although reaching overall higher recall values for the ensemble of simulated data (Figure 3a; mean recall ± standard deviation = 0.826 ± 0.053 vs. 0.816 ± 0.237 for mapQ filtering vs. alignment identity filtering, respectively). Similar results were observed for precision, which increases with the stringency of the mapQ threshold reaching overall high values for the ensemble of simulated data (Figure 3b; mean precision standard deviation = 0.95 ± 0.089 vs. 0.65 ± 0.24 for mapQ filtering vs. alignment identity filtering, respectively). When both filtering strategies were compared in terms of F1 scores, defined as the harmonic mean of precision and recall (high F1 scores being a good trade-off between the two metrics [37]), we observed that the filtering by mapQ produces significantly higher F1 scores than filtering by alignment identity under all community compositions regarding species richness (Figure 3c; *p*-value < 0.05; Wilcoxon rank-sum test). 

Finally, in terms of similarity between estimated species abundance and the reference values, we observed different results for different complexities of simulated communities. In low-richness samples (R50, R150), the similarities increased with the mapQ threshold from 5 to 30, but this was not the case for more complex samples (R250–R450), where the similarities did not improve as the mapQ threshold was increased further than mapQ = 5 (Figure 3d). On the contrary, the similarity of simulated abundances with the reference abundances significantly decreased with the stringency of mapQ filtering in simulated samples with 450 species (Appendix A). Pairwise comparison of F1 scores in species richness estimates and Spearman rhos of pairwise similarities with reference abundances across simulations with different mapQ filter thresholds shows that both metrics were strongly associated (Appendix A), which suggests that the overall accuracy of species richness estimates determines how well the simulated abundance profiles resemble the reference abundance data. These results show that for complex microbiome communities, like those of the human gut, a mapQ threshold of 5 gave the best results in terms of species richness estimations (based on F1 scores) and similarity regarding the estimated relative abundance profiles with the reference data (Appendix A). This parameter, however, should be adapted depending on the estimated complexity of the target microbial community.

### 3.3. Validation of the Bioinformatics Pipeline with ZymoBIOMICS Mock Community

The quantification of the ZymoBIOMICS mock community combining Centrifuge taxonomic binning and filtering by minimap2 alignment of read bins vs. the corresponding reference genomes with parameters derived from simulation experiments (primary alignments only, minimum mapQ = 5) reproduced the composition of the mock community with high accuracy and reduced the number of miss-assignments in comparison with the classification based on Centrifuge only (Appendix A). This also led to a higher overall similarity of microbiome composition (estimated as 1-Bray–Curtis beta-diversity) with the reference mock community with the combination of Centrifuge and minimap2 filtering (0.91) than with raw Centrifuge results (0.88). 

### 3.4. DNA Extraction Kits Influenced Read Length Distribution

When testing DNA extraction kits on dry spoon stool samples from healthy volunteers, all kits except the “ZimoBiomics DNA Mini Kit” (Ozyme) provided sufficient DNA quantity and quality for sequencing. Thus, we examined the library preparation and sequenced all kits, except the Ozyme kit. We observed a bimodal distribution of read lengths across the Invitrogen and Mo Bio DNA library preparation kits, with a high proportion of long reads (>1.1 kb), whereas with the Promega and Qiagen kits, the distribution was skewed towards smaller reads (Figure 4a). The Mo Bio kit led to the production of sequences with a mean of 8.5 kb in size, while the Invitrogen kit produced sequences up to 24 kb. The Qiagen kit and the two Promega (Promega 1 and Promega 2) kits yielded sequences up to 17 kb but with a higher proportion of short reads. The fraction of classified sequences was significantly higher for long reads (log2-length > 9.96; *p*-value = 7.5 × 10^−9^; Wilcoxon signed-rank test), with on average 39% of long reads successfully classified after the two-step’s procedure based on Centrifuge compared with 24% for shorter reads (log2-length < 9.96 (Figure 4d)), confirming initial observations with simulated data about the importance of this parameter in the taxonomic classification of ONT reads. The examination of alpha diversity also showed significant differences by DNA extraction kit, with high diversity levels in Invitrogen samples (Figure 4e, *p*-value = 0.0061, Kruskal–Wallis test; *p*-value = 9.4 × 10^−4^ Invitrogen vs. Promega 1 and Promega 2 kits; post hoc pairwise Dunn’s test). Finally, we observed that the differences between the microbiome compositions of the different replicates were mainly explained by the collection day (R^2^ = 0.45, *p*-value = 0.001), followed by the sample donor (R^2^ = 0.03, *p*-value = 0.001) and DNA extraction kit (R^2^ = 0.02; *p*-value = 0.001) (Figure 4h; PERMANOVA test, marginal effects on a multivariate model with collection day, DNA extraction kit, sample donor, DNA fragmentation, and DNA end repair). Based on these observations, the Invitrogen kit was selected as the preferred extraction kit for sequencing.

### 3.5. DNA Fragmentation and End Repair 

The first step in ONT’s library preparation protocol is DNA fragmentation to generate 8 kb fragments [27]. Using three different samples from one subject, DNA fragmentation had no effect on read length distribution (Figure 4b), species richness (Figure 4f; *p*-value > 0.05, Wilcoxon rank-sum test), or microbiome composition based on PCoA ordination (Figure 4i; R^2^ = 0.001, *p*-value = 0.949, PERMANOVA test). Therefore, we decided to exclude this step from our experimentation framework. 

The ONT DNA preparation protocol recommends DNA-end repair. We evaluated the effect of DNA-end repair on read length and microbial diversity by extracting DNA from stools of the same three subjects using the Invitrogen kit and excluding the DNA fragmentation step. We found similar profiles of read length distributions between samples with end-repair and no end-repair steps (Figure 4c), no significant impact on species richness (Figure 4g; *p*-value > 0.05, Wilcoxon rank-sum test), and no significant impact on microbiome composition (Figure 4j; R^2^ = 0.001, *p*-value = 0.877, PERMANOVA test). This step was then omitted from our proposed protocol.

### 3.6. Optimized DNA Extraction Protocol Improved ONT Read Length and Microbial Diversity Estimation

Based on the simulated and real data, read length was an important parameter to maximize the efficiency of the taxonomic classification of ONT reads. Consequently, we aimed at increasing the read length by optimizing DNA extraction and library preparation using the Invitrogen kit. We followed recommendations from the International Human Microbiome Standards (IHMS) consortium [20] (Figure 5A), but to increase the proportion of long reads, we improved the sequencing library preparation protocol by modifying two main steps. The first one was the “end-prep” step, which prepares the binding of the adapter to the DNA after two incubation periods. We used the “NEBNext Ultra II End Repair/dA-Tailing Module” from New England Biolabs (NEB). In the ONT protocol, “end-prep” reaction incubating is recommended for 5 min at 20 °C, followed by 5 min at 65 °C. However, the NEB kit recommends a first incubation at 20 °C for 30 min, followed by a second incubation at 65 °C for 30 min. Given the lack of effects of the ONT end-prep protocol, we attempted end repair using the NEB kit and recommended protocol (Figure 5B). 

In the library preparation step, DNA was purified by using “Agencourt AMPure XP” beads (Beckman Coulter, Roissy CDG, France), which use solid-phase reversible immobilization (SPRI) paramagnetic bead technology, which selectively binds nucleic acids according to type and size. Agencourt AMPure XP utilizes an optimized buffer, polyethylene glycol (PEG), to selectively bind DNA fragments. The size of the fragments eluted from the beads is determined by PEG concentration. For example, if 50 μL of beads is added to a 50 μL DNA sample, a SPRI/DNA ratio of 1 is obtained. When this ratio was changed, the length of the fragments binding and/or remaining in the solution also changed. The SPRI/DNA ratio was disproportionately associated with the DNA fragment size, which is due to the fragment size affecting the total charge carried by the molecule. Thus, long DNA fragments would have a greater proportion of negative charges, which promotes their electrostatic interaction with the beads and allows a priority link to the carboxyl molecules. The ONT protocol was developed based on the DNA fragmentation of 8 kb sequence length, and the SPRI/DNA ratio must be equal to 1. In order to promote the selection of larger DNA fragments by paramagnetic beads, we reduced the SPRI/DNA ratio to 0.4 (Figure 5B). The chosen ratio was based on the SPRI technology documentation [38] and ONT users’ recommendations from the “Community” forum [27]. 

Thus, we performed two modifications (end-prep and DNA purification) with the Invitrogen extraction protocol, referred to as “Optimized Invitrogen”. This optimization step was performed for six samples from one healthy subject (from the MetaCardis study), collected at six time points. Each sample was extracted using the standard “Invitrogen” protocol and with the “optimized” protocol. DNA yields extracted from this optimized protocol were five times greater than the ones obtained with the standard kit (55 ng vs. 300 ng, *p*-value < 0.0001). The ratio of the absorbance at 260/230 was higher with the optimized protocol, 2.11 vs. 1.38, respectively (*p*-value = 0.0007), and the absorbance ratio at 260/280 significantly increased, 1.89 vs. 1.73, in the nonoptimized one, respectively (*p*-value = 0.0046) (Appendix A). 

The read length was also improved (Figure 6a). The standard Invitrogen protocol produced two populations of reads with average read lengths of 500 and 6000 bp, while the optimized protocol produced a single read population with an average read length of 6000 bp. According to this, we observed a significant increase in the fraction of classified reads in comparison with the fraction obtained with the initial protocol recommended by Invitrogen (Figure 6b; 29.72% with optimized protocol vs. 23.92% with original protocol; *p*-value = 0.031, Wilcoxon signed-rank test). We observed an increased microbial diversity (observed species) in four of the six samples with the optimized protocol even if this variation remains insignificant (*p*-value = 0.31; Wilcoxon signed-rank test) (Figure 6c). Finally, PCoA (Figure 6d) showed that differences in microbiome composition across samples are explained mainly by the collection date of the samples (R^2^ = 0.89, *p*-value = 0.001, PERMANOVA test), with no significant effect of the extraction kit on the overall microbiome composition (R^2^ = 0.026, *p*-value = 0.906, PERMANOVA test). Altogether, the optimized DNA extraction protocol exhibited a better DNA yield and purity, longer sequences than the usual protocol, leading to a significant yield improvement (fraction of classified reads) of the taxonomic binning with no impact on the overall microbiome composition of the samples. 

### 3.7. Impact of Stool Sampling and Storage on Sequence Length and Diversity

Subjects’ stool samples were initially collected in a dry spoon tube and rapidly frozen at −80 °C to ensure the stability of the bacterial DNA. However, an increasing number of sampling systems contain a solution that can stabilize bacterial DNA at room temperature for periods ranging from 60 days (DNA Genotek) up to 2 years (NORGEN Biotek and Ozyme). We evaluated the effects of room temperature (RT) stabilized samples on bacterial DNA extraction, library preparation, and sequencing. We prepared six DNA libraries from stools of 12 healthy subjects collected by different protocols in three different stabilizing kits: “Omnigen Gut for Microbiome” (DNA Genotek), “Stool Nucleic Acid Collection and Preservation Tubes” (Norgen Biotek), and “DNA/RNA Shield-Fecal Collection Tube” (Ozyme). Regarding read lengths, we observed similar unimodal distribution towards long reads across all experiments (Appendix A). We did not observe significant differences between the three collection kits in the fraction of classified reads (Appendix A; *p*-value = 0.38 in −80° group; *p*-value = 0.28 in RT group). Additionally, we observed no significant differences in species richness (Appendix A; *p*-value = 0.71 in −80° group, *p*-value = 0.41 in RT group; Kruskal–Wallis test), although we could notice a tendency with Norgen and Omnigen kits to decrease microbial diversity at room temperature in comparison with −80 °C storage (Appendix A). In contrast, we observed significant variations in microbial diversity by donor (Appendix A; *p*-value = 0.0017, Kruskal–Wallis test), being the variable with the highest impact on microbiome composition by PERMANOVA analyses (R^2^ = 0.82; *p*-value = 0.001) in comparison with the collection kit (R^2^ = 0.11, *p*-value = 0.009) and storage conditions (R^2^ = 0.07; *p*-value = 0.029) (Appendix A). Thus, sampling methods at room temperature with DNA stabilization performed similarly to snap-frozen samples, and the choice of the sampling kit might depend on practical feasibility sampling for the subject and kit price. We chose the Ozyme kit for its practicality for users to collect stool samples and its relative costs.

### 3.8. Optimized ONT Protocol Compared with Illumina SOLiD Sequencing

We compared ONT-obtained QM profiles with those generated with other sequencing technologies from the Microbaria study [8]. We selected 33 presurgery samples covering the extremes of microbiome diversity defined as microbial gene richness (13 samples from individuals with high gene count (HGC) and 20 samples from individuals with low gene count (LGC)). ONT abundance profiles were generated using the two bioinformatics workflows described in the methods section, based on Centrifuge and mapping over the IGC. DNA from 21 of the 33 samples were also extracted with the optimized Invitrogen protocol and sequenced using Illumina technology. Semi-quantitative metagenomic profiles from Illumina samples were generated by mapping reads against the IGC as described in [9].

First, we compared the estimates of microbial diversity from ONT (gene richness from IGC mapping and observed species from Centrifuge classification) and Illumina sequencing (gene richness from IGC mapping) with the gene richness inferred from the original SOLiD sequencing of these samples. SOLiD sequencing generated 4.38 × 10^7^ single reads of 35 bases (sd = 1.86 × 10^7^) per sample on average, representing 1.53 × 10^9^ base pairs overall (sd = 6.52 × 10^8^). With the ONT, we generated an average of 1.53 × 10^5^ reads per sample between 200 bp and 24 kb (sd = 6.07 × 10^4^), representing 4.19 × 10^8^ bp overall (sd = 1.607 × 10^8^).

We observed significant positive associations between Centrifuge-based diversity estimates from ONT sequencing and the reference gene richness from SOLiD sequencing based on the IGC (Spearman rho = 0.59, *p*-value = 3 × 10^−4^ for observed species based on Centrifuge results; Figure 7a). These similarities increased with the use of the IGC as reference database for diversity estimations (Spearman rho = 0.74, *p*-value = 2 × 10^−6^ for gene richness based on ONT read mapping over the IGC; Figure 7b). However, the similarity was higher with gene richness estimates based on Illumina sequencing despite differences in library preparation (Spearman rho = 0.86, *p*-value < 2.2 × 10^−6^; Figure 7c). When we integrated the scaled diversity profiles (dividing each diversity estimate by the maximum value in each sequencing source for these 33 samples (ranges from 0 to 1)) and ordered them based on the reference gene richness in the original publication [8], we observed that DNA extraction had an impact on diversity. Both ONT and Illumina sequencing using the same DNA extraction method showed similar variations in microbial diversity estimates compared with the reference SOLiD data (Figure 7d). This included a switch in the sample showing the highest diversity (i.e., MB12 sample with Illumina and ONT sequencing, MB21 sample with SOLiD sequencing; Figure 7d). 

The high similarity between ONT and Illumina datasets was confirmed in an ordination framework, where we integrated the genus-level abundance profiles from IGC quantification with the three sequencing technologies (ONT, Illumina, SOLiD), where we observed that sample products of the same DNA extraction method (Illumina and ONT) are closer in a PCoA ordination (Appendix A) and in hierarchical clustering analyses (Appendix A).

### 3.9. ONT Pipeline Detects Target Species and Functional Profiles

Regarding taxonomic feature quantification, we found a good agreement between ONT sequencing and both Illumina and SOLiD sequencing data. Based on Centrifuge, we observed a positive correlation of relative abundances in 91 of the 95 common taxonomic features with SOLiD quantifications (96%; mean Spearman rho ± standard deviation = 0.68 ± 0.15 (species), 0.58 ± 0.32 (genus), 0.53 ± 0.3 (family), 0.5 ± 0.26 (order), 0.51 ± 0.24 (class), 0.62 ± 0.18 (phylum)). A total of 72 of these features (76%) were significantly associated (FDR < 0.05, Spearman correlations). Similarly, we observed a positive correlation in 94 of the 101 common taxonomic features with Illumina quantification (93%; mean Spearman rho ± standard deviation = 0.74 ± 0.22 (species), 0.62 ± 0.26 (genus), 0.63 ± 0.27 (family), 0.63 ± 0.24 (order), 0.66 ± 0.2 (class), 0.68 ± 0.22 (phylum)). A total of 78 of these features (77%) were significantly associated (FDR < 0.05, Spearman correlations) (Figure 7e). 

Using the quantification of metagenomic species (MGS) based on ONT mapping over the IGC, the relative abundances of 137 common taxonomic features with SOLiD quantifications were positively associated (mean Spearman rho ± standard deviation = 0.68 ± 0.12 (species), 0.6 ± 0.16 (genus), 0.63 ± 0.17 (family), 0.58 ± 0.2 (order), 0.59 ± 0.2 (class), 0.58 ± 0.16 (phylum)), 128 of which (93%) were significantly associated (FDR < 0.05, Spearman correlations). A similar comparison with MGS relative abundance products of Illumina sequencing gave 133 of the 137 common taxonomic features positively associated (98%, mean Spearman rho ± standard deviation = 0.77 ± 0.14 (species), 0.75 ± 0.19 (genus), 0.72 ± 0.21 (family), 0.65 ± 0.22 (order), 0.67 ± 0.2 (class), 0.64 ± 0.26 (phylum)), 122 of which (90%) were significantly associated (FDR < 0.05, Spearman correlations) (Appendix A). 

Importantly, these results also showed that the similarities in the relative abundances of taxonomic features between ONT and Illumina quantifications were significantly higher than between ONT and SOLiD sequencing (Appendix A; *p*-value < 0.05 for comparisons at the species and genus levels with ONT Centrifuge results; *p*-value < 0.005 for comparisons at the species, genus, and family levels with ONT IGC results).

We made similar observations with functional profiles based on KEGG modules. Using Centrifuge, 76% and 72% of the functional modules were positively associated with the equivalent modules quantified with Illumina and SOLiD sequencing, respectively, whereas this fraction substantially increased to 98% and 98% with ONT abundance data based on IGC quantifications (Appendix A). This difference may be related to the different contents of both genomic reference spaces (Centrifuge genomes and IGC), which can have a major impact on the quantification of functional modules if differences in database composition also result in differences in gene content. Importantly, we observed that DNA extraction also had an impact on the similarity between functional profiles, with ONT functional profiles being more similar to Illumina functional profiles based on both Centrifuge (*p*-value = 0.0046 Wilcoxon rank-sum tests of Spearman rho distributions between ONT–SOLiD comparisons and ONT–Illumina comparisons; mean Spearman rho ± standard deviation = 0.24 ± 0.32 (ONT Centrifuge vs. Illumina IGC functional module abundances), 0.20 ± 0.28 (ONT Centrifuge vs. SOLiD IGC functional module abundances)) and IGC quantifications (*p*-value = 0.0046 Wilcoxon rank-sum tests of Spearman rho distributions between ONT–SOLiD comparisons and ONT–Illumina comparisons; mean Spearman rho ± standard deviation = 0.53 ± 0.22 (ONT IGC vs. Illumina IGC functional module abundances), 0.44 ± 0.18 (ONT IGC vs. SOLiD IGC functional module abundances)) (Appendix A).

Finally, we reproduced with ONT data previously reported associations between functional modules and microbiome diversity at similar strength as with Illumina and SOLiD data. We found significant positive associations between the sporulation module md:M00485 (KinABCDE-Spo0FA (sporulation control) two-component regulatory system) and microbial diversity (Appendix A), which was in agreement with estimations of 50–60% of bacteria from the gut microbiome of healthy individuals producing resilient spores, being a basic feature of the human microbiome with a key impact in bacterial persistence and the spread of microbes between individuals [39]. This was also the case for the negative association between modules involved in the biosynthesis of bacterial lipopolysaccharide (LPS) and microbial diversity (Appendix A), in line with the association of obesity and other metabolic disorders with an increase in blood LPS concentration [40].

## 4. Discussion

Here, we presented a novel protocol and analytical pipeline enabling the quantification of the gut microbiome features using Oxford Nanopore Technologies. This technology supports easy access and use of high-throughput sequencing at competitive costs as well as fast data production and analyses of the results. We believe that this protocol enables the study of gut microbiome samples in the context of clinical applications or group studies. We improved the protocols for both the wet-lab (from DNA extraction to sequencing) and the data analysis. We also compared results with second-generation sequencing methods (Illumina and SOLiD) in a previously described patient cohort. This was driven by (1) an initial assessment of the best parameters in terms of alignment of ONT sequencing reads from simulated metagenomic datasets with different levels of complexity and (2) the development of a bioinformatics pipeline, which combines rapid k-mer-based classification of ONT reads with read alignments vs. reference genomes to improve the quantification of microbiome species diversity and composition. This also included the taxonomic and functional profiling of ONT metagenomic data from reference genomes (Centrifuge approach) and gut microbiome nonredundant gene catalogs. Previous studies have proposed similar approaches based on Centrifuge for the real-time metagenomic profiling from ONT data [13] and more specifically for the metagenomic profiling of fecal and oral swabs [41], but not allowing functional profiling or metagenomic profiling using nonredundant gut microbiome gene catalogs that maximize the genomic knowledge of the gut microbiome ecosystem. The simulation experiments revealed that filtering strategies commonly used with second-generation sequencing technologies, such as high sequence identity thresholds, could not be extrapolated to highly error-prone reads, such as those produced by ONT. In contrast, the mapping quality based on Nanopore-adapted sequence aligners such as minimap2 showed significantly better performance in terms of precision and recall of species richness composition estimates at different complexities of simulated communities. MapQ scores of 5 gave the best results regarding estimates of species richness and relative abundances of microbial species, being particularly suitable for complex ecosystems such as the human gut. 

Regarding sample processing, we elaborated a DNA extraction protocol from human stools that provide high DNA quality. Studies over the past years have used bacterial DNA or RNA to explore microbial communities in diverse ecosystems, including stool samples from large cohorts [3,42,43]. Authors have used different DNA extraction protocols and different sequencing techniques (Illumina, SOLiD, Ion Proton). Multiple studies have also noted “batch” [44] effects and differences in data analyses [45,46], which introduce biases in analytical comparisons. Thus, the need for procedure standardization has been highlighted by several reports, as illustrated by the IHMS consortium [47]. They compared 21 DNA extraction methods using whole-genome metagenomic shotgun sequencing with Illumina HiSeq 2000 technology and assessed the taxonomic profile and functional variability while standardizing the stages of stool collection, bacterial DNA stabilization, library preparation, and sequencing. This resulted in the generation of recommendations that would improve DNA extraction in terms of yield and quality. 

Taking into account IHMS recommendations, we further optimized the microbial DNA extraction protocol, which showed DNA yield improvement. We worked on two critical steps, bacterial wall lysis and protein/RNA elimination. 

Sampling conditions, storage, and harmonization have also been shown to be critical in affecting microbiome results. Although storing fecal samples at 4 °C appeared to protect bacterial DNA from degradation, a reduction in microbial diversity was observed [48]. A previous study showed that prior storage of stool samples at 4 °C (1 h) before placing them at −20 °C had a large impact on the taxonomic composition at the genus and species levels [49]. However, these studies were conducted before the development and widespread use of commercially available fecal collection kits with stabilizing solution. Here, our results suggest that sample storage temperature is not a significant factor as long as guidelines from manufacturers are followed. The effect of sample storage kit type or temperature on sequencing and microbiome results is largely outweighed by interdonor variation.

Since we identified read length as a critical criterion for subsequent bioinformatics analyses, we also improved the library preparation protocol to increase the proportion of long reads by optimizing the end-prep and DNA purification steps in the library. Finally, the PCoA of different wet-lab experiments showed that individual microbiome composition drove most of the variation observed in microbial diversity and semi-quantitative metagenomic profiles obtained from ONT sequencing data, with no apparent batch effects associated to different wet-lab steps (DNA fragmentation, DNA end-repair, collection kits, DNA library preparation, or sequencing run).

To examine the relevance of our pipeline in human cohorts, we performed a comparison of the results obtained with ONT sequencing with those obtained with SOLiD and Illumina sequencing on human stool samples collected in the “Microbaria” study [9]. For gene richness, microbiome composition, and functional modules, the similarity was higher between ONT and Illumina sequencing compared with SOLID. ONT and Illumina sequences were generated from the same DNA extracted with the optimized protocol, which emphasizes the importance of DNA extraction protocols in semi-quantitative metagenomic profiles.

The low throughput of ONT sequencing is, however, one of its major drawbacks for quantitative metagenomic studies of complex microbial ecosystems, such as the human gut microbiome. Nevertheless, the results showed a high similarity in bacterial diversity estimation between the two sequencing methods in our test samples. Despite improvements in experimental protocol, the demultiplexing step needs to be improved. For instance, the ratio of unclassified reads was about 25%, knowing that the sequencing depth of ONT was low, and the error rate elevated. However, the low classification rates (25%) did not seem to have an impact on the bacterial diversity and the bacterial composition estimates. In this context, we could consider ONT in the context of semi-quantitative metagenomic studies as a “shallow-sequencing” method in the line of proposed low-sequencing depth approaches to characterize microbial ecosystems more accurately than 16S barcoding approaches and with lower costs than deep shotgun sequencing [50].

## 5. Conclusions

Nanopore-based technology is proposed as easily accessible due to relatively low costs and a small benchtop footprint, providing an avenue to perform NGS in clinical settings. Admittedly, this technology has some drawbacks, such as relatively modest sequencing depth and error rates that remain high (2–5%) [51] compared with Illumina (0.1%) [52]. Through accurate assessment of experimental and bioinformatics steps, our current work demonstrates that this technology is suitable to carry out semi-quantitative metagenomic studies in the human gut microbiome. The maximization of ONT read lengths by our experimental protocol, in addition to being key in maximizing the efficiency of the taxonomic binning (fraction of classified reads), could be of major importance in additional aspects of metagenomic analyses, such as de novo assembly of microbial genomes and the improvement of the genomic completion of metagenomic species (MGS) derived from human gut microbiome gene catalogs. Our bioinformatics pipeline also extends for the first time the scope of metagenomic profiling of ONT reads to these gene catalogs, which has been of pivotal importance as reference genomic spaces used in multiple semi-quantitative metagenomic studies, opening the way for the validation of disease biomarkers derived from these studies in clinical practice. In this context, we show that ONT consistently replicated results obtained with other sequencing technologies for intestinal microbiome diversity and the composition of main phyla in patients with severe obesity. This proposed workflow paves the way to taxonomic and functional profiling of microbial communities with this sequencing technology at competitive costs and fast data, which corresponds to a great need in the microbiome community.

## Figures and Tables

**Figure 1 genes-12-01496-f001:**
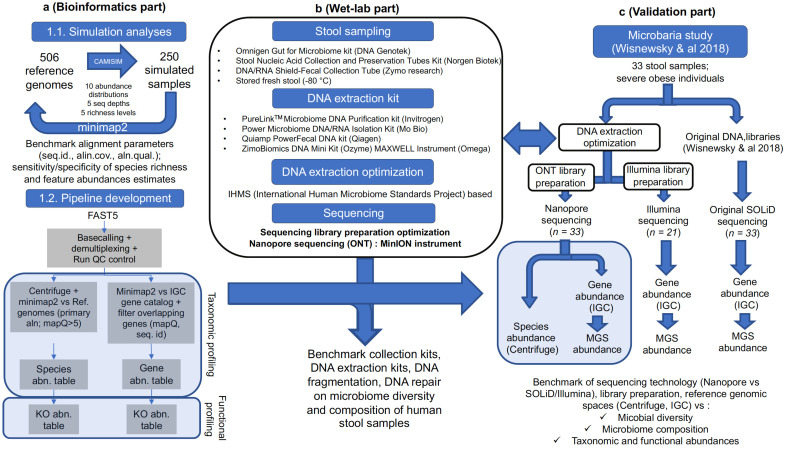
Summary of the workflow: (**a**) Simulated data processing. (**b**) Wet-lab optimization. (**c**) Summary of ONT sequencing comparison with Illumina and SOLiD technologies. IGC: Integrated Gene Catalog, MGS: metagenomic species, ONT: Oxford Nanopore Technologies.

**Figure 2 genes-12-01496-f002:**
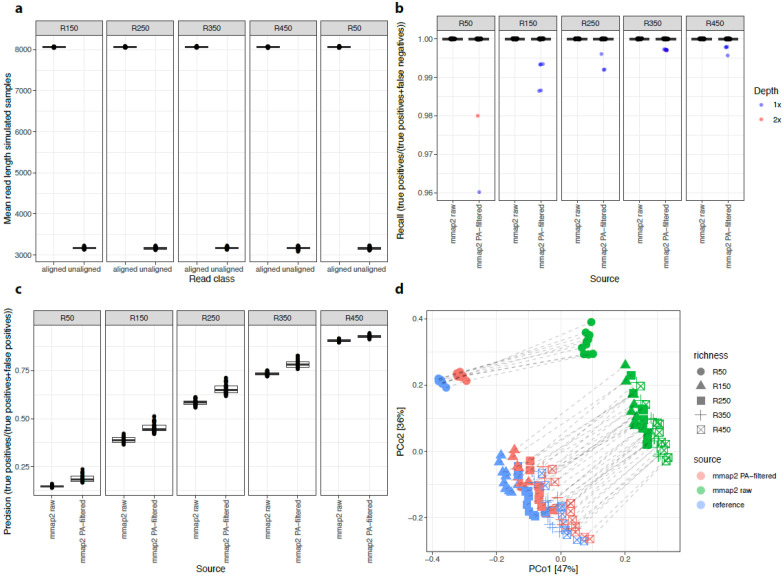
Metagenomic profiles from simulated samples between minimap2 results and minimap2 results filtered from secondary alignments. (**a**) Boxplots of mean lengths of ONT reads of 250 simulated samples (y-axis) between those aligned and unaligned over the 506 reference genomes from minimap2 results (x-axis). (**b**) Boxplots of recall values of species richness estimates in 250 simulated samples (y-axis) between metagenomic profiles inferred from all minimap2 alignments (mmap2 raw) and from minimap2 primary alignments only (mmap2APfilt, x-axis). For simulated samples not reaching the recall of 1, the sequencing depth is highlighted in different colors. (**c**) Boxplots of precision values of species richness estimates in 250 simulated samples (y-axis) between metagenomic profiles inferred from all minimap2 alignments (mmap2 raw) and from minimap2 primary alignments only (mmap2APfilt, x-axis). (**d**) PCoA of metagenomic profiles from the reference and 250 simulated samples inferred from all minimap2 alignments (mmap2raw) and from minimap2 primary alignments only (mmap2APfilt, x-axis). Dashed lines connect points coming from the same sample (reference, simulated ones; 3 points per sample).

**Figure 3 genes-12-01496-f003:**
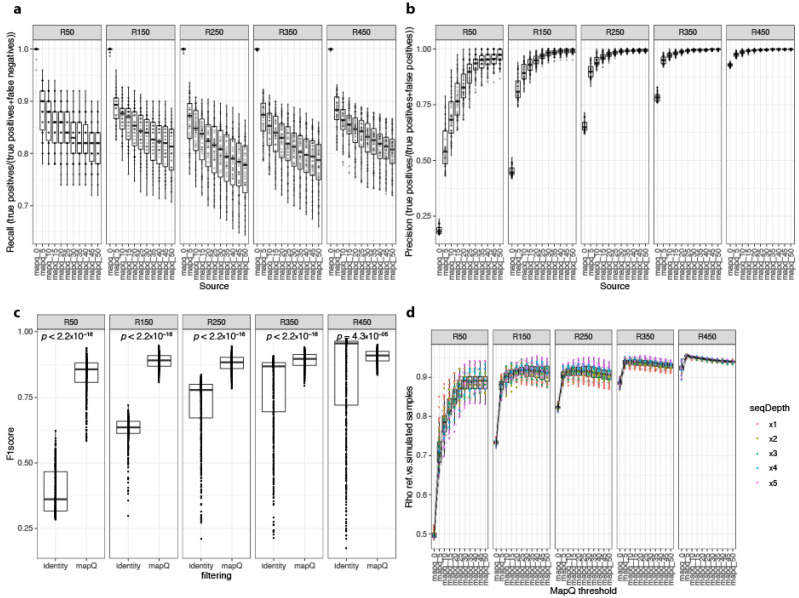
Impact of filtering minimap2 primary alignments of Nanopore reads at different thresholds of mapQ score. Boxplots of recall (**a**) and precision (**b**) values of species richness estimates in 250 simulated samples (y-axis) between metagenomic profiles inferred from primary alignments of ONT reads filtered by different thresholds of mapQ score (from 0 to 50; x-axis) stratified by the number of species in reference metagenomic profiles. (**c**) Boxplots of F1 scores (harmonic mean of precision and recall) in species richness estimates between simulated datasets filtered by alignment identity and mapQ scores stratified by the number of species in reference metagenomic profiles. *p*-Values of product of Wilcoxon rank-sum tests are shown for each pairwise comparison. (**d**) Boxplots of Spearman rho coefficients in correlation analyses between taxonomic profiles of reference and simulated samples (y-axis) at different thresholds of sequence identity (from 0% to 90%; x-axis) stratified by the number of species in reference metagenomic profiles. Points are colored according to the sequencing depth of simulated samples.

**Figure 4 genes-12-01496-f004:**
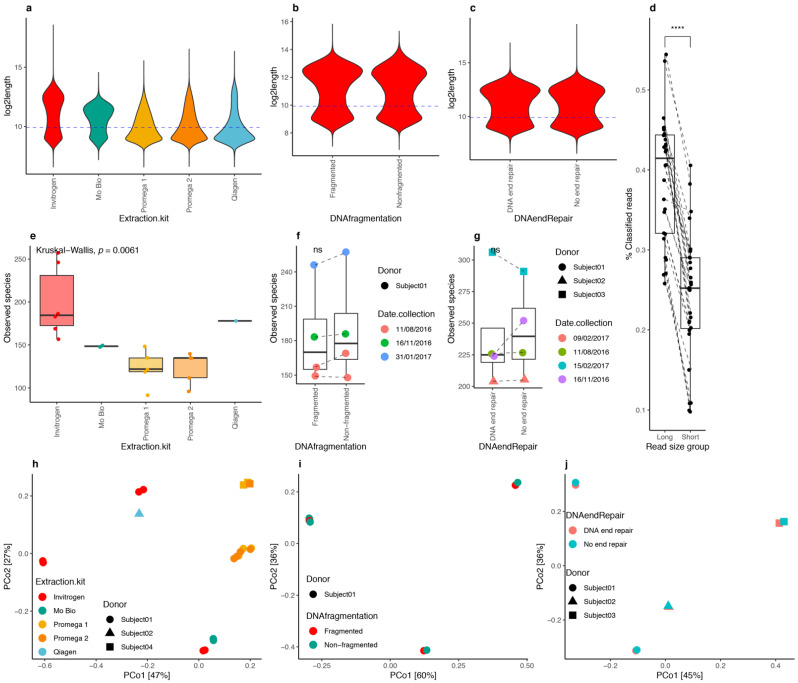
DNA extraction kits, fragmentation, and end-repair impact on human stool metagenomic composition from ONT sequencing data. Read length distributions of ONT reads across different DNA extraction kits ((**a**) *n* = 29) and between DNA fragmentation ((**b**) *n* = 6 paired samples fragmented/nonfragmented) and DNA end repair ((**c**) *n* = 6 paired samples end vs. no end repair) steps for Invitrogen samples. Blue dashed lines correspond to the median value of log2-transformed read lengths used to stratify reads as long or short. (**d**) Differences between the fraction of classified reads by the Centrifuge approach between long and short reads for 29 samples in panel (**a**). (**e**) Differences in microbial diversity (observed species) between extraction kits (*n* = 29). (**f**) Differences in microbial diversity (observed species) by DNA fragmentation (*n* = 4 paired samples). (**g**) Differences in microbial diversity (observed species) by DNA end-repair step (*n* = 4 paired samples). (**h**) PCoA ordination of 29 samples in panel (**a**) colored by extraction kit. (**i**) PCoA ordination of 8 samples in panel (**f**). (**j**) PCoA of 8 samples of panel (**g**) colored by DNA end-repair step. ns (panel (**f**,**g**)) = nonsignificant differences in paired Wilcoxon rank-sum tests. **** = *p*-value < 0.0001 in paired Wilcoxon rank-sum test.

**Figure 5 genes-12-01496-f005:**
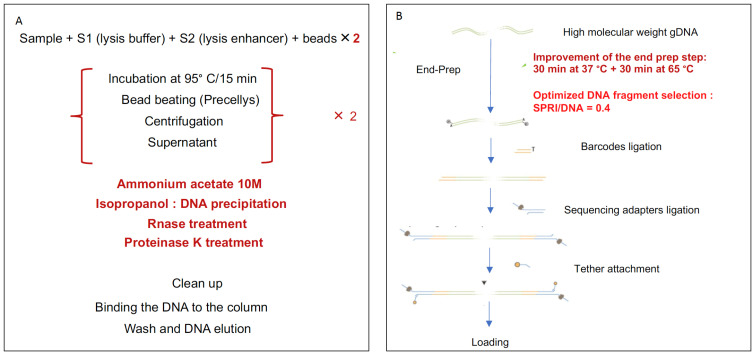
Optimization of DNA extraction and library preparation protocols. (**A**) Steps of the bacterial DNA extraction protocol. In black, the steps include the protocol of the Invitrogen kit, in red, the improvement steps recommended by the IHMS consortium. (**B**) Improvement of library preparation by the application of NEB recommendation and decrease in the SPRI/DNA ratio.

**Figure 6 genes-12-01496-f006:**
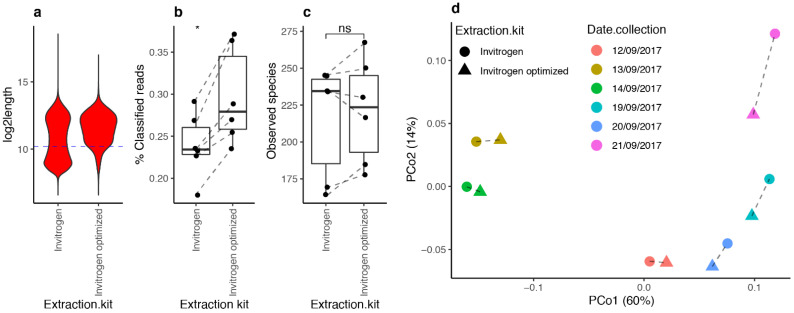
Impact of Invitrogen optimized protocol on human stool metagenomic composition from Nanopore sequencing data. (**a**) Read length distributions of ONT reads in *n* = 6 samples extracted with original (Invitrogen) and optimized (Invitrogen optimized) protocols. Blue dashed lines correspond to the median value of log2-transformed read lengths used to stratify reads as long or short. (**b**) Differences in the fraction of classified reads between protocols (*n* = 6 paired samples; * = *p*-value < 0.05, paired Wilcoxon rank-sum test). (**c**) Differences in microbial diversity between protocols (*n* = 6 paired samples; ns = *p*-value > 0.05, paired Wilcoxon rank-sum test). (**d**) PCoA ordination from genus-level beta-diversity matrix (Bray–Curtis) of 12 samples extracted with Invitrogen optimized kit and original Invitrogen kit. Dashed lines connect samples coming from the same fecal stool sample collected at different dates.

**Figure 7 genes-12-01496-f007:**
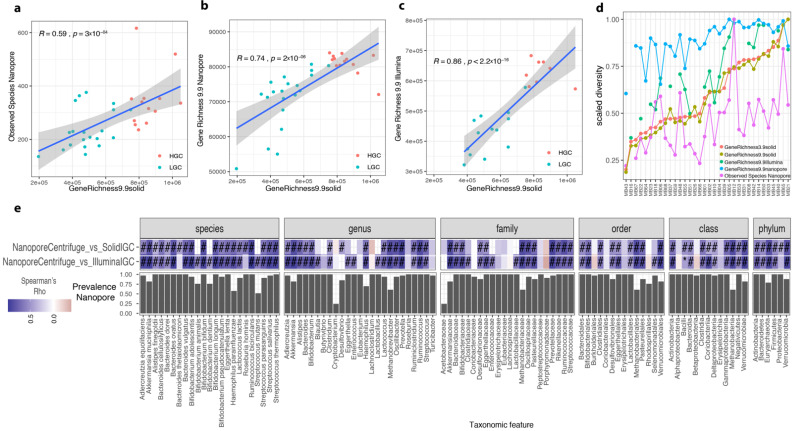
Comparison of semi-quantitative metagenomic profiles of Microbaria samples between sequencing technologies. Correlation between gene richness from SOLiD sequencing (x-axis) and observed species inferred from Nanopore (ONT) sequencing data using Centrifuge approach ((**a**) *n* = 33), gene richness inferred from ONT sequencing data ((**b**) *n* = 33), and gene richness inferred from Illumina sequencing data ((**c**) *n* = 21). The strength of the similarities was evaluated with Spearman correlation test (Spearman rho and *p*-value included in the scatterplots). (**d**) Line plots representing the scaled diversity (from 0 to 1) of Microbaria samples from different diversity metrics based on SOLiD, ONT, and Illumina sequencing data. Samples in x-axis are ordered based on the scaled diversity of the gene richness from the original Microbaria study (GeneRichness3.9SOLiD). (**e**) Heatmap of Spearman rho representing similarities in abundance vectors of taxonomic features in x-axis between ONT quantifications based on Centrifuge data and Illumina and SOLiD quantifications based on metagenomic species of the IGC (y-axis; #= *p*-value adj < 0.05, BH method; * = *p*-value < 0.05). On the bottom of the heatmap is represented the prevalence of taxonomic features in x-axis based on ONT sequencing data.

## Data Availability

Sequences have been deposited in the European Bioinformatics Institute (EBI) European Nucleotide Archive (ENA) under study accessions PRJEB47445 (Nanopore and Illumina sequences) and PRJEB23292 (SOLiD sequences). The computational pipeline is freely available at https://git.ummisco.fr/pipelines/nanopore on 22 September 2021. Other data are available on request.

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
