# Peer review of "Exploring Semi-Quantitative Metagenomic Studies Using Oxford Nanopore Sequencing: A Computational and Experimental Protocol"

_genes, 2021, doi:10.3390/genes12101496_

Round 1

Reviewer 1 Report

Abstract: correct "remains A challenges".

The term "quantitative metagenomics" appears somewhat disputable since the introduction of protocols like QMP (quantitative microbiome profiling; DOI: 10.1038/nature24460): the latter provides truly quantitative results (bacterial cells concentrations per taxon) by augmenting WGS with flow cytometry or qPCR. Without total cell count enumeration, the former remains rather SEMIquantitative.

lines 45-46: Please specify if you are considering taxonomic or functional microbial richness or both.

56: reSSources.
Consider applying auto-spellchecker for the whole text. Remove doubled spaces, too. And there are misprints in the figures.

113: IGC - please expand.

115-116: How can one calculate profiles of species from catalog of genes? Did you mean MGS? Please explain.

Consider reducing the number of figures (too many - particularly, too many box/violin plots) by moving most to supplementary materials.

Author Response

First of all, thank you for taking the time to read our work and for your constructive comments and remarks. We have tried to respond to your comments as well as possible and have had the manuscript proofread by our native American colleague for English. Below are the responses to your comments:

- The term "quantitative metagenomics" appears somewhat disputable since the introduction of protocols like QMP (quantitative microbiome profiling; DOI: 10.1038/nature24460): the latter provides truly quantitative results (bacterial cells concentrations per taxon) by augmenting WGS with flow cytometry or qPCR. Without total cell count enumeration, the former remains rather SEMIquantitative.

We replaced “quantitative” by “semi-quantitative” in the manuscript.

- lines 45-46: Please specify if you are considering taxonomic or functional microbial richness or both.

- The study referred in the mentioned lines considers microbial gene richness derived from metagenomic read mapping over non-redundant gene catalogs of the gut microbiome. Microbial richness has been replaced by microbial gene richness to address reviewer comment

- 56 : reSSources.
Consider applying auto-spellchecker for the whole text. Remove doubled spaces, too. And there are misprints in the figures.

The text has been fully reviewed to address reviewer’s comment

- 113 : IGC - please expand.

IGC means Integrated Gene Catalog of the human gut microbiome. The abbreviation is introduced at his first appearance in the manuscript (line 104-105).

115-116: How can one calculate profiles of species from catalog of genes? Did you mean MGS? Please explain.

The reviewer is correct about the way taxonomic profiles are computed from the catalog of genes (by means of the metagenomic species). The way MGS abundances are computed from gene abundances are described in the lines 150-153 of the manuscript.

- Consider reducing the number of figures (too many - particularly, too many box/violin plots) by moving most to supplementary materials.

two figures have been moved in the supplemental data.

Reviewer 2 Report

Here the authors developed a computational and experimental protocol for whole genome quantitative metagenomics of human gut microbiome with Oxford Nanopore sequencing technology.

The work is overall well done and written.

Please consider the following minor comments:

  • Figure 1b should be revised. The authors should differentiate the first two boxes (stool sampling is repeated twice). Furthermore, under DNA extraction optimization, the authors also report library preparation optimization, etc. Please, rearrange and define abbreviations in the legend (e.g., IGC, MGS).
  • Abbreviations should be defined at their first occurrence.
  • Accession number. The authors must provide the link for Reviewers to access sequences.

Author Response

First of all, thank you for taking the time to read our work and for your constructive comments and remarks. We have tried to respond to your comments as well as possible and have had the manuscript proofread by our native American colleague for English. Below are the responses to your comments:

- Figure 1b should be revised. The authors should differentiate the first two boxes (stool sampling is repeated twice). Furthermore, under DNA extraction optimization, the authors also report library preparation optimization, etc. Please, rearrange and define abbreviations in the legend (e.g., IGC, MGS).

This figure was corrected.

- Abbreviations should be defined at their first occurrence.

It was done

- Accession number. The authors must provide the link for Reviewers to access sequences.

The sequences has been deposited in the ENA repository under the study accession PRJEB47445. Data Availability Statement section of the manuscript has been updated accordingly (lines 789-792).